# Phase-specific premotor inhibition modulates leech rhythmic motor output

Martina Radice[1,2], Agustin Sanchez Merlinsky[1], Federico Yulita[1,3], Lidia Szczupak[1,2]*

[1]Instituto de Fisiología, Biología Molecular y Neurociencias (IFIBYNE-UBA-CONICET), Buenos Aires, Argentina; [2]Departamento de Fisiología, Biología Molecular y Celular, Facultad de Ciencias Exactas y Naturales, Universidad de Buenos Aires, Buenos Aires, Argentina; [3]Departamento de Física, Facultad de Ciencias Exactas y Naturales, Universidad de Buenos Aires, Buenos Aires, Argentina

## eLife Assessment

The medicinal leech preparation is an amenable system in which to understand the neural basis of locomotion. Here, a previously identified non-spiking neuron was studied in leech and found to alter the mean firing frequency of a crawl-related motoneuron, which fires during the contraction phase of crawling. The findings are **valuable**, and the experiments were diligently done and considered **solid**. The results lay a foundation for additional studies in this system.

*For correspondence:
lidiaszczupak@gmail.com

Competing interest: The authors declare that no competing interests exist.

## Abstract

Understanding how motoneuron activity is finely tuned remains an open question. Leeches are a highly suitable organism for studying motor control due to their well-characterized behaviors and relatively simple nervous system. On solid surfaces, leeches display crawling, a rhythmic motor pattern that can be elicited in the isolated nerve cord or even in single ganglia isolated from it. This study aimed to learn how this motor output is shaped by concurrent premotor signals. Specifically, we analyzed how electrophysiological manipulation of a premotor nonspiking (NS) neuron, which forms a recurrent inhibitory circuit (analogous to that formed by vertebrate Renshaw cells), shapes the leech crawling motor pattern. The study included a quantitative analysis of putative motor units active throughout the fictive crawling cycle that shows that the rhythmic motor output in isolated ganglia mirrors the phase relationships observed in vivo. Taken together, the study reveals that the premotor NS neurons, under the control of the segmental pattern generator, modulated the degree of excitation of motoneurons during crawling in a phase-specific manner.

## Introduction

Animal locomotion involves the coordination of multiple muscles under the control of the nervous system. To ensure smooth movements, diverse regulatory mechanisms operate at different hierarchical levels within the motor system, narrowing down the degrees of freedom of a highly distributed system with multiple units. The overall functional organization and operational logic of motor systems, comprising the components at all levels of the hierarchy and their feedforward and feedback interactions, remain, to a great extent, an open question in both vertebrates and invertebrates (*Kiehn, 2016*; *Arber and Costa, 2018*; *Barkan and Zornik, 2019*; *Grätsch et al., 2019*); this includes the mechanisms that operate at the level of the nerve cord (*Grillner and El Manira, 2020*; *McLean and Dougherty, 2015*; *Gal et al., 2017*; *Sengupta and Bagnall, 2023*; *Calabrese and Marder, 2025*). Analysis of this organization across different organisms, developmental stages, and behavioral repertoires

**Figure 1.** The crawling motor pattern. (**A**) Schematic description of a leech crawling step that results from coordinated waves of elongation (i–ii) and contraction (iii–iv) phases, anchored on front and rear suckers. (**B**) Intracellular recording of CV and extracellular recording of DE-3 (in DP nerve) motoneurons during a dopamine-induced crawling episode in an isolated midbody ganglion. Recording diagram on the left. (**C**) Putative neuronal circuitry underlying crawling and the recurrent inhibitory circuit in a midbody ganglion. Units C and E correspond to contraction and elongation units of the segmental oscillator, respectively. DE-3 and CV are examples of motoneurons involved in each phase. NS neuron is connected to DE-3 and CV (**Rela and Szczupak, 2003**; **Rodriguez et al., 2009**) through chemical and electrical synapses; the + and –symbols indicate the polarity at which the rectifying synapse conducts.

has provided fruitful information. Leeches have been highly useful to study motor control due to their robust repertoire of motor behaviors, which are executed by a relatively simple body plan and controlled by a similarly simple nervous system. This system is composed of a chain of 21 midbody ganglia flanked by head and tail brains (**Wagenaar, 2015**), where each midbody ganglion contains the sensory and motor neurons that innervate the corresponding segment (**Muller et al., 1981**).

On solid surfaces, leeches exhibit a robust rhythmic motor pattern known as crawling (**Figure 1A**). This behavior results from waves of elongation and contraction of the body along its antero-posterior axis, as the animal is anchored on the posterior and anterior suckers, respectively (**Stern-Tomlinson et al., 1986**, **Figure 1A**). Fictive crawling (*crawling*) can be monitored in the isolated nervous system, where identified interneurons and motoneurons can be recorded intracellularly and extracellularly (**Baader, 1997**; **Baader and Kristan, 1992**; **Eisenhart et al., 2000**). The pattern is characterized by the alternating activation of the motoneurons, such as CV and DE-3, that innervate circular and longitudinal muscles, respectively (**Figure 1B**). This pattern can be readily evoked by dopamine in the isolated nerve cords (**Puhl and Mesce, 2008**), in short chains of ganglia (**Kearney et al., 2022**), and in single isolated ganglia (**Puhl and Mesce, 2008**; **Rodriguez et al., 2012**). These studies established

that each ganglion contains the network responsible for producing the rhythmic motoneuron activity compatible with crawling. Note that in the context of this article fictive crawling refers to the rhythm generation but does not include the connectivity that rules intersegmental interactions and the consequent intersegmental lags.

Studies on motor behaviors in diverse organisms show that, at the final stage of neural processing, motoneuron activity is controlled by multiple signal sources (*El Manira, 2023*; *Zhen and Samuel, 2015*; *Rotstein et al., 2017*). Analysis of the nature and effects of these signals on the motor pattern is an active field of research. The present work has focused on the role played by nonspiking (NS) premotor neurons on leech crawling. These neurons are present as bilateral pairs in each segmental ganglion and are functionally analogous to mammalian Renshaw cells (*Szczupak, 2014*). These spinal cord cells deliver inhibitory signals to the motoneurons in proportion to the motoneuron activity, forming an activity-dependent feedback mechanism that regulates motoneuron output (*Alvarez and Fyffe, 2007*).

Each pair of NS neurons is at the center of a recurrent inhibitory circuit mediated by chemical and electrical synaptic connections with the motoneurons (*Figure 1C*; *Rela and Szczupak, 2003*; *Rodriguez et al., 2009*). In the context of this circuit, the activity of excitatory motoneurons evokes chemically mediated inhibitory synaptic potentials in NS. Additionally, the NS neurons are electrically coupled to virtually all excitatory motoneurons via rectifying junctions that conduct when the transjunction NS-motoneuron potential is negative. In physiological conditions, this coupling favors the transmission of inhibitory signals from NS to motoneurons. Given this recurrent inhibitory circuit, NS premotor neurons could operate as modulators of motoneuron activity. A previous study (*Rodriguez et al., 2012*) indicated that NS membrane potential oscillates in tune with the *crawling* motor pattern and that this premotor neuron can indeed modulate motor activity. In that study *crawling* was monitored by the recording of a particular motoneuron, the dorsal excitor cell 3 (DE-3). However, the wide connectivity of NS with several excitatory motoneurons that fire during *crawling* called for a more comprehensive readout of the motor pattern.

The present study aimed to evaluate the influence of the premotor NS neuron across the different stages of *crawling*. For this purpose, we describe the pattern of activity of different neurons during *crawling* that were simultaneously monitored through extracellular nerve recordings while manipulating the NS membrane potential. The study reveals that the premotor neuron mediated a reduction of the firing frequency of motoneurons recruited during a specific phase of *crawling*.

## Results

### NS membrane potential oscillations during *crawling*

To study the role that premotor NS neurons play in fictive crawling (*crawling*), we first analyzed the correspondence between NS electrophysiological activity and the motoneuron output during this motor pattern. Since motoneuron DE-3 (dorsal excitor cell 3), which innervates the longitudinal dorsal muscle fibers, is the most robust proxy of *crawling* (*Puhl and Mesce, 2008*), extracellular recordings of this motoneuron in the DP nerve were combined with NS intracellular recordings. As shown previously (*Rodriguez et al., 2012*), the NS membrane potential showed rhythmic oscillations tuned to DE-3 activity, with hyperpolarizations occurring in conjunction with DE-3 bursts and depolarizations during DE-3 silent phases (*Figure 2A*). Here, a cross-correlation analysis between NS and DE-3 firing frequency revealed a marked anticorrelation, with a cross-correlation index of –0.61±0.03 (mean ± SEM) at a lag close to 0 s (0.24±0.10 s) (*Figure 2B*). This result indicates that NS hyperpolarization takes place in phase with DE-3 bursts.

Within the framework of the recurrent inhibitory circuit, NS is the target of inhibitory signals evoked by excitatory motoneurons via chemically-mediated polysynaptic pathways (*Rela and Szczupak, 2003*, *Figure 1C*). Thus, it was hypothesized that the NS hyperpolarizations observed during *crawling* resulted from inputs originating in DE-3 and other motoneurons that fire in phase with it (*Rodriguez et al., 2012*). If this hypothesis was correct, the onset of DE-3 bursts should precede the onset of NS hyperpolarization. Because the cross-correlation was performed on binned DE-3 firing frequency curves, the peak lag reported does not offer a proper estimation of the relative timing between the two signals. To gain a more precise assessment, NS intracellular recordings were segmented into individual cycles and aligned with the first spike of the corresponding DE-3 burst (*Figure 2C*). As

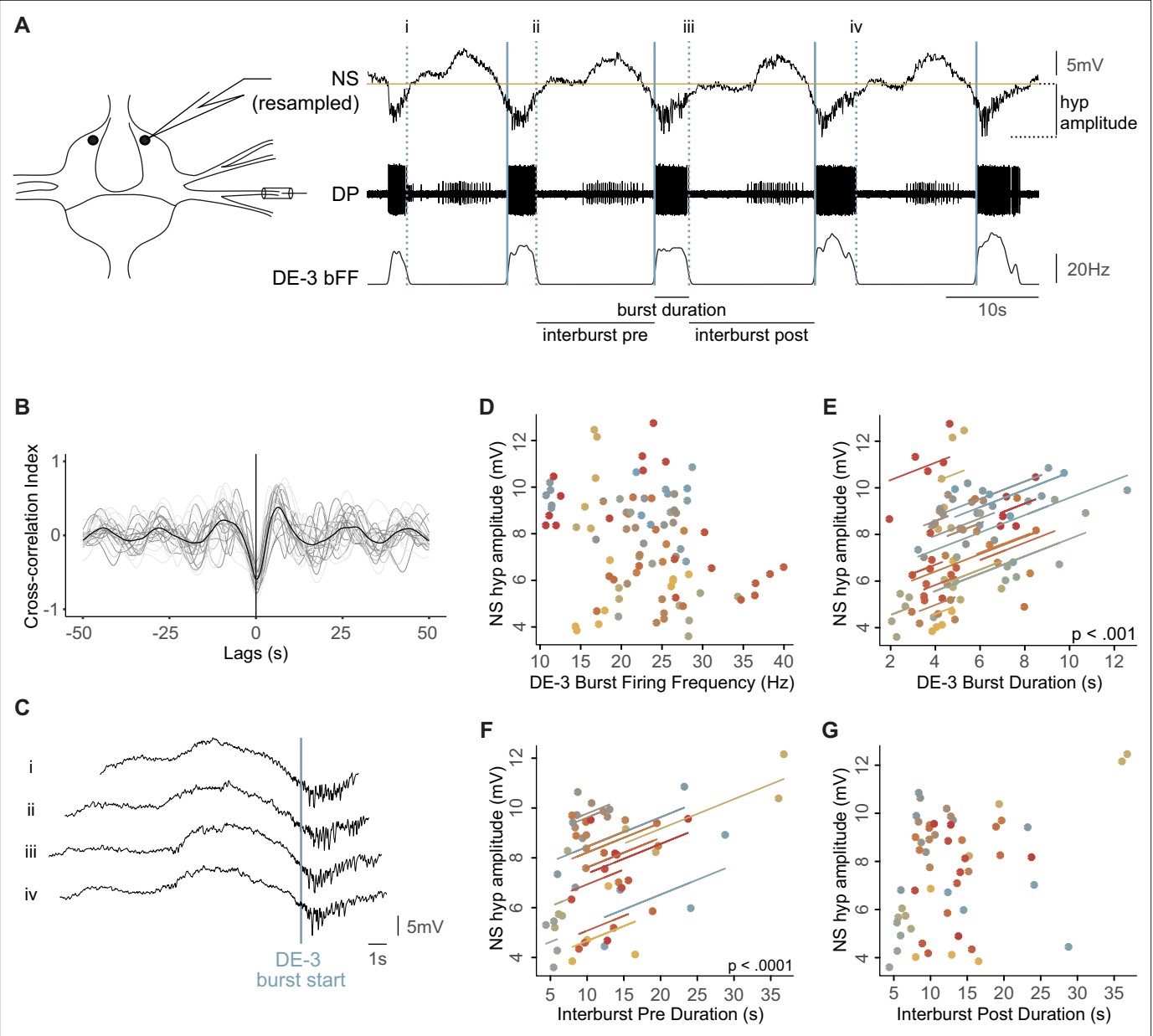

**Figure 2.** Nonspiking (NS) activity correlates with the *crawling* motor pattern. (**A**) NS intracellular recording, DP extracellular recording, and DE-3 binned firing frequency (bFF) during a crawling episode. NS recording was resampled to match the data rate of DE-3 bFF. Recording diagram on the left. The blue vertical line marks the beginning and the blue dotted line the end of each DE-3 burst. Different crawling features and NS hyperpolarization amplitude (NS hyp amplitude) are indicated. (**B**) Cross-correlograms of the DE-3 bFF curves and NS resampled traces. Each gray trace corresponds to an individual experiment. The black curve corresponds to the mean cross-correlogram. (n=23 crawling episodes containing 5 cycles, obtained in 23 ganglia from 18 leeches). (**C**) Successive cycles of the NS recording shown in panel (**A**), segmented at the times indicated by the dotted lines and aligned by the times indicated by solid blue lines indicated in (**A**). (**D**) NS hyperpolarization amplitude as a function of DE-3 Burst FF. (**E**) Like in (**D**), as a function of DE-3 burst duration. (**F**) Like in (**D**), as a function of interburst lag preceding DE-3 bursts. (**G**) Like in (**D**), as a function of the interburst lag after DE-3 bursts. For (**D–G**), n=19 crawling episodes, with five cycles each. In (**D–G**), the correlation was tested using linear mixed-effects models, contemplating different experiments as a random variable. In (**D**) and (**G**), p>0.05, and in (**E**) and (**F**), p<0.05. The lines correspond to regressions obtained from the linear mixed-effects model. The coefficients were 0.374±0.089 mv/s for (**E**) and 0.122±0.034 mv/s for (**F**).

shown in the figure, because of the NS membrane potential oscillations, it is not possible to establish a sharp onset for the NS hyperpolarizing response. However, the example illustrates that the onset of the DE-3 bursts occurred after NS hyperpolarization was already initiated. These findings rule out the possibility that the NS hyperpolarization was originated by DE-3 bursting. Instead, they suggest that

NS and DE-3 received concomitant rhythmic inputs of opposite sign—that is, hyperpolarizing inputs in NS and depolarizing ones in DE-3.

To further analyze the putative concomitant inputs from the pattern generator onto NS and DE-3, we evaluated the correlation between different features of their rhythmic activity (*Figure 2A*). We found no relationship between the amplitude of NS hyperpolarization and DE-3 mean firing frequency (mFF) (*Figure 2D*). However, surprisingly, the former exhibited a high correlation with DE-3 burst duration (*Figure 2E*). This result led to examining how the amplitude of NS hyperpolarization related to the time between consecutive DE-3 bursts (interbursts time), preceding (pre) and following (post) the NS hyperpolarization (*Figure 2A*). The data show a significant correlation of the amplitude of NS hyperpolarization with the pre-interburst duration, but not with the post-interburst duration (*Figure 2F and G*). Taken into consideration the putative model presented in *Figure 1C*, in which the oscillator units C and E inhibit each other, the longer the activity of unit E, the larger the rebound in unit C and its downstream effects on NS hyperpolarization and on DE-3 burst duration.

These results indicate that, in addition to the known recurrent inhibitory circuit between NS and motoneurons, NS neurons receive an inhibitory signal during *crawling*, which is in phase with DE-3 activity. This inhibitory signal seems to have originated from the same source that activates DE-3 motoneurons. In that case, it seems puzzling to find a lack of correlation between NS hyperpolarization amplitude and DE-3 firing frequency. However, this observation might be explained by considering that DE-3 is subjected to two counteracting effects at the same time: an excitatory input from the oscillator unit C and an inhibitory input from NS via the rectifying electrical junctions (*Figure 1C*).

## Effect of NS on DE-3 activity during *crawling*

In a previous study (*Rodriguez et al., 2012*), we showed that hyperpolarization of NS during *crawling* reduced the firing frequency of DE-3, suggesting that the premotor neuron preserved the influence that exhibits in basal conditions during the rhythmic motor pattern (*Rodriguez et al., 2009*). Here we aimed at testing whether the inhibitory signal that NS receives during crawling is indeed transmitted to the motoneurons. For this purpose, the premotor neuron was transiently removed from the circuit. In line with the properties of the rectifying electrical synapse between NS and the motoneurons, depolarizing NS neurons is equivalent to disconnecting them from the network (*Rela and Szczupak, 2007*). The work by *Rodriguez et al., 2012* showed that NS depolarization increased DE-3 firing frequency, but the study lacked appropriate controls that we carried out here.

Depolarizing steps applied in NS neurons in the course of *crawling* episodes (*Figure 3A*) shifted their membrane potential beyond +100 mV, well above the average membrane potential of the motoneurons. An important control introduced here evaluates potential drifts in *crawling* features over time. For this purpose, we compared two series of experiments: one in which NS was subjected to depolarizing steps (depo) and a control (ctrl) in which it was not. We assessed changes in cycle period, DE-3 mFF, burst duration, and duty cycle for both treatments (depo, ctrl) across different epochs (pre, test, post).

Transient NS depolarizations had no effect on the *crawling* period (measured in end-to-end cycles, see 'Materials and methods'), DE-3 burst duration and duty cycle (*Figure 3B*), but produced a 60% increase in DE-3 mFF (*Figure 3B and C*).

In the control series, the analysis of *crawling* features over time shows that cycle period and DE-3 burst duration decreased over epochs, while the duty cycle remained constant. Similar results were obtained when cycles were segmented from start-to-start (data not shown). Thus, throughout the dopamine perfusion, the motor pattern exhibited an acceleration, but the duty cycle remained stable.

As predicted based on the known circuit (*Figure 1C*), this series of experiments indicates that inhibitory signals onto NS premotor neurons are transmitted to DE-3 motoneurons, thereby limiting their firing frequency by counteracting their excitatory drive during *crawling*.

## A more comprehensive view of *crawling*

As indicated above, the activity of motoneuron DE-3 has been widely used as a proxy of *crawling*. However, focusing on a single motoneuron neglects possible effects of NS on other phases of the motor pattern. To evaluate whether the effect of the premotor NS neuron extends to other motoneurons active during *crawling*, we performed extracellular recordings of DP along with AA and/or PP root nerves. Single units were identified using a spike sorting algorithm (*Figure 4Ai–Aii*) that enabled

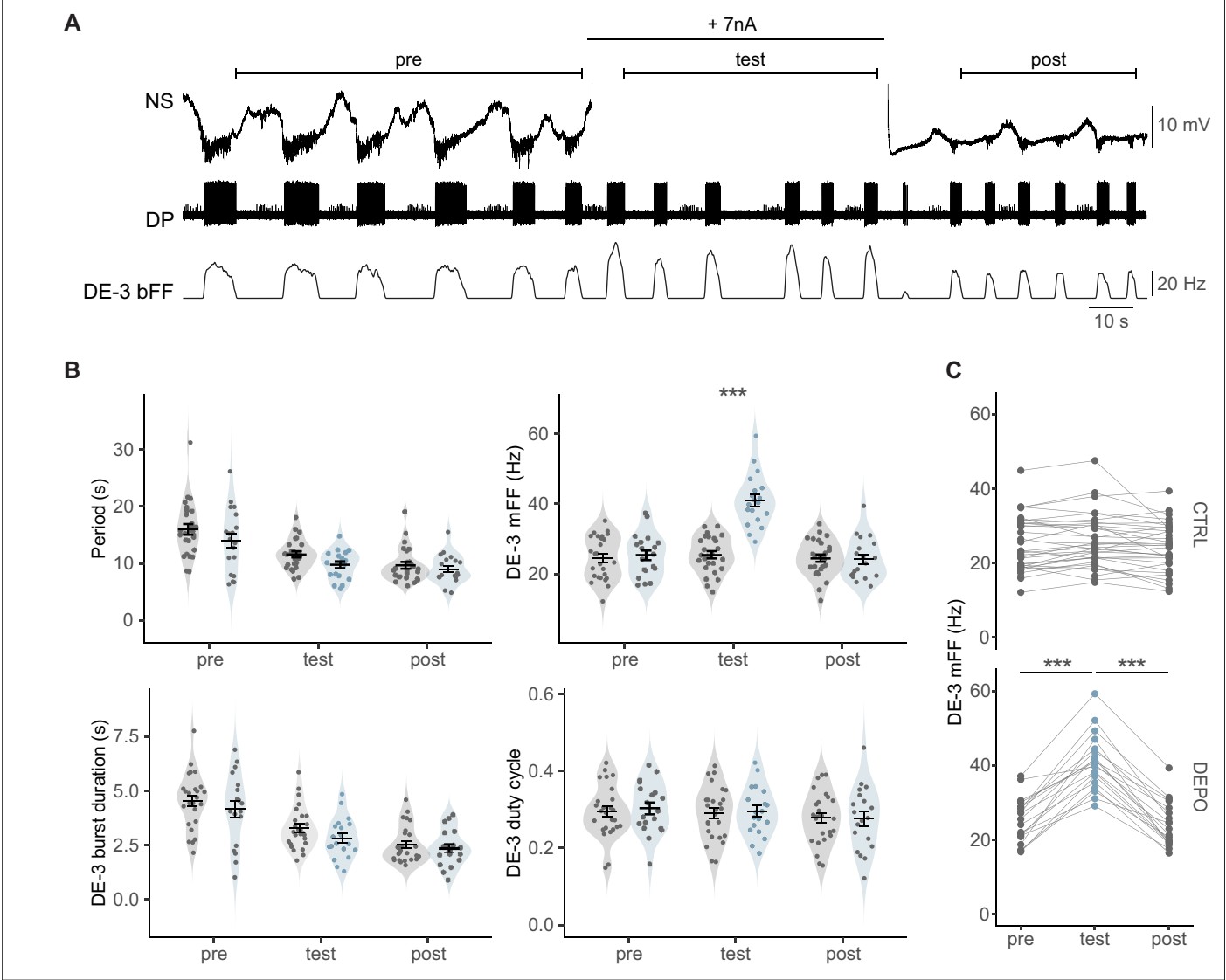

**Figure 3.** Effect of nonspiking (NS) on *crawling*. (**A**) Intracellular NS and extracellular DP recordings during a dopamine-induced crawling episode. A+7 nA pulse was injected in NS at the time indicated by the thick line. Segments above the recordings mark three epochs: previous (pre), during (test), and after (post) the pulse. (**B**) Crawling period (end-to-end), DE-3 mFF, burst duration, and duty cycle for cycles in the three epochs marked in (**A**), for control experiments where no current was injected in NS (gray violins, n=120 cycles from 26 episodes in 26 ganglia from 16 leeches) and for experiments where NS was depolarized (light blue violins, n=92 cycles from 19 episodes in 19 ganglia from 15 leeches); each point corresponds to the mean value during the epoch for each individual experiment. Bars provide the mean ± SEM. ***p<0.001 (interaction between epoch and treatment factors). (**C**) Mean DE-3 mFF for the three epochs from each individual experiment for control (CTRL) and depolarizing (DEPO) tests. Lines link dots that correspond to each individual experiment. ***p<0.001 (pairwise simple contrasts, Δ pre-test=–15.591 ± 1.037 Hz and Δ test-post=16.648 ± 0.919 Hz).

a systematic analysis of their activity. The firing pattern of detected units (*Figure 4Aiii*) was expressed as binned firing frequency (bFF). Because DE-3 is used as a reference marker of the cycle, segmentation of the traces was performed using the start-to-start approach (see 'Materials and methods'). This rhythmic activity shows that some units fire during the contraction stage, defined by DE-3 activity, while others fire during the elongation phase.

Plotting the bFF of the different rhythmic units as a function of phase within a *crawling* cycle (start-to-start) allowed for the manual classification of the unit. Based on their activity profile relative to DE-3, three types were identified: in-phase, anti-phase, or in-phase-early-onset (*Figure 4B*). The in-phase units fire simultaneously with DE-3, the anti-phase units fire in between DE-3 bursts, and the in-phase-early-onset units are active in phase with DE-3 but start firing before DE-3 burst onset (*Figure 4C*).

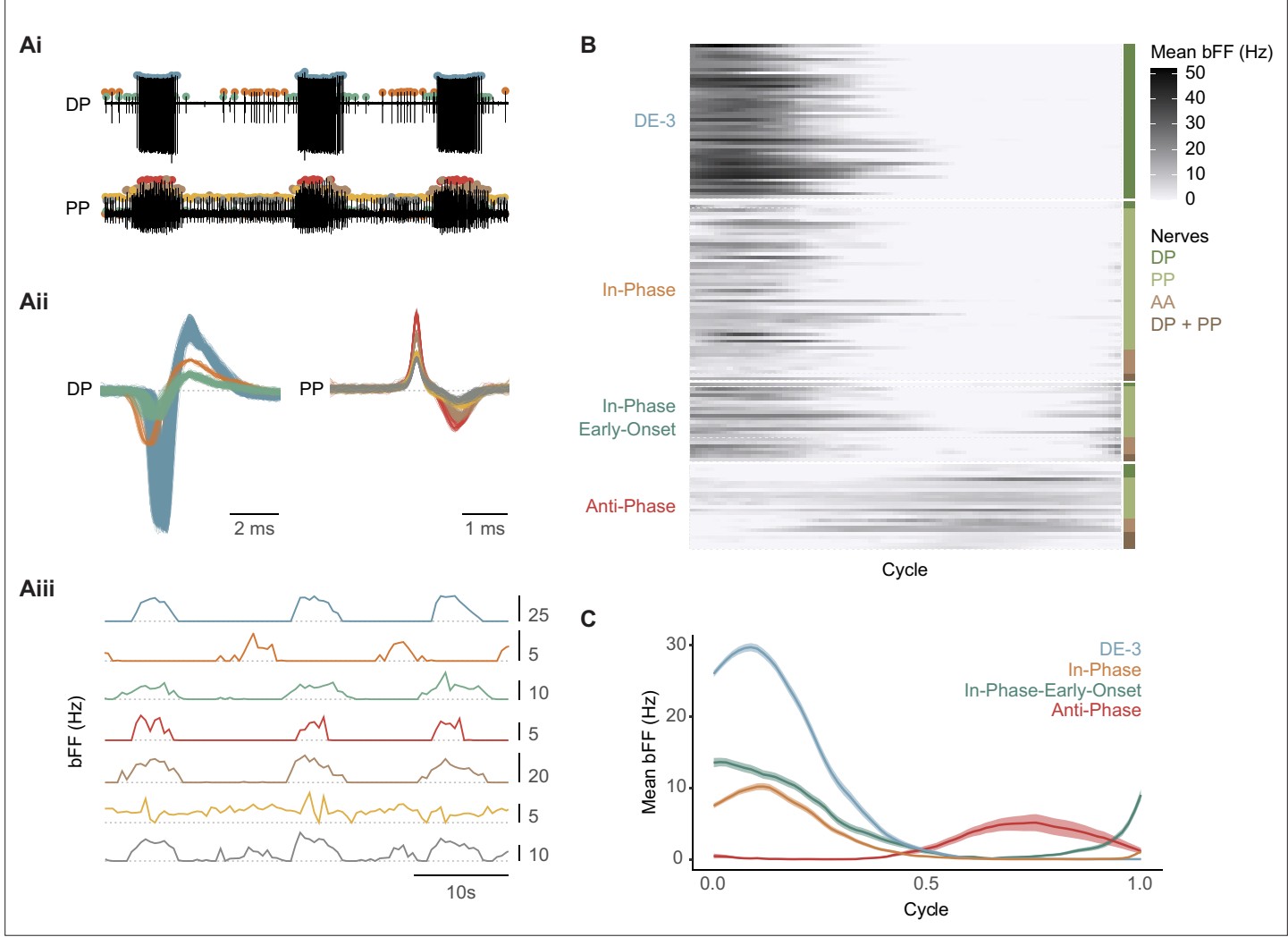

**Figure 4.** Profile of motoneuron activity during *crawling*. (**Ai**) Representative example of a spike sorting analysis implemented in paired extracellular recordings of DP and PP nerves during a crawling episode. Each color dot singles out a distinct active unit across the recordings. (**Aii**) Spikes waveforms corresponding to spikes detected in the DP and PP nerves in Ai. (**Aiii**) Binned firing frequency (bFF) calculated for each unit identified in (**A**). The horizontal dotted lines indicate bFF = 0 Hz. Scales on the right are in Hz. (**B**) Raster plot showing mean bFF across a cycle, set from the first spike of a DE-3 burst to the first spike in the next burst, obtained from 147 rhythmic units identified in DP, PP, and AA nerves (n=46 ganglia in 24 animals). The values are the mean of four crawling cycles for each unit. The units were classified into four groups, indicated on the left. The nerves from which the motoneuron activity was recorded are indicated on the right. The gray scale on the right indicates the bFF value in Hz. (**C**) Mean bFF for all the units included in each group. The shaded area represents the standard error of the mean.

Multiple intracellular recordings represent a challenge, and therefore recording the activity of different motoneurons during crawling has been restricted to one (*Puhl and Mesce, 2008*) or two (*Rodriguez et al., 2012*) cells at a time. While multiple extracellular recordings have been performed previously (*Eisenhart et al., 2000*), these results (*Figure 4*) present the first quantitative analysis of multiple units activated throughout the *crawling* cycle in this type of recordings. Nerve roots carry both sensory and motor axons; however, the units recorded during *crawling* in isolated ganglia are more likely to be motoneurons. Although the preparation is detached from peripheral inputs, sensory neurons could still receive inputs from the segmental pattern generator. The recorded nerves conduct axons from mechanosensory pressure (P) and touch (T) neurons (*Nicholls and Baylor, 1968*). However, P cells show no activity during dopamine-induced *crawling*, and T cells receive inhibitory inputs in phase with DE-3 bursts (*Alonso et al., 2020*). Axons from stretch receptors also run through the recorded nerves, but these neurons transmit information passively (*Blackshaw and Thompson, 1988*; *Cang and Friesen, 2000*). Therefore, the in-phase units are expected to be motoneurons that control

the contraction stage by exciting or inhibiting the longitudinal or circular muscles, respectively, and the anti-phase units motoneurons that control the elongation stage, by exciting or inhibiting the circular or longitudinal muscles, respectively. It is plausible that in-phase-early-onset are inhibitors of circular muscles, preparing the conditions for a fast contraction stage. Further studies should confirm this hypothesis.

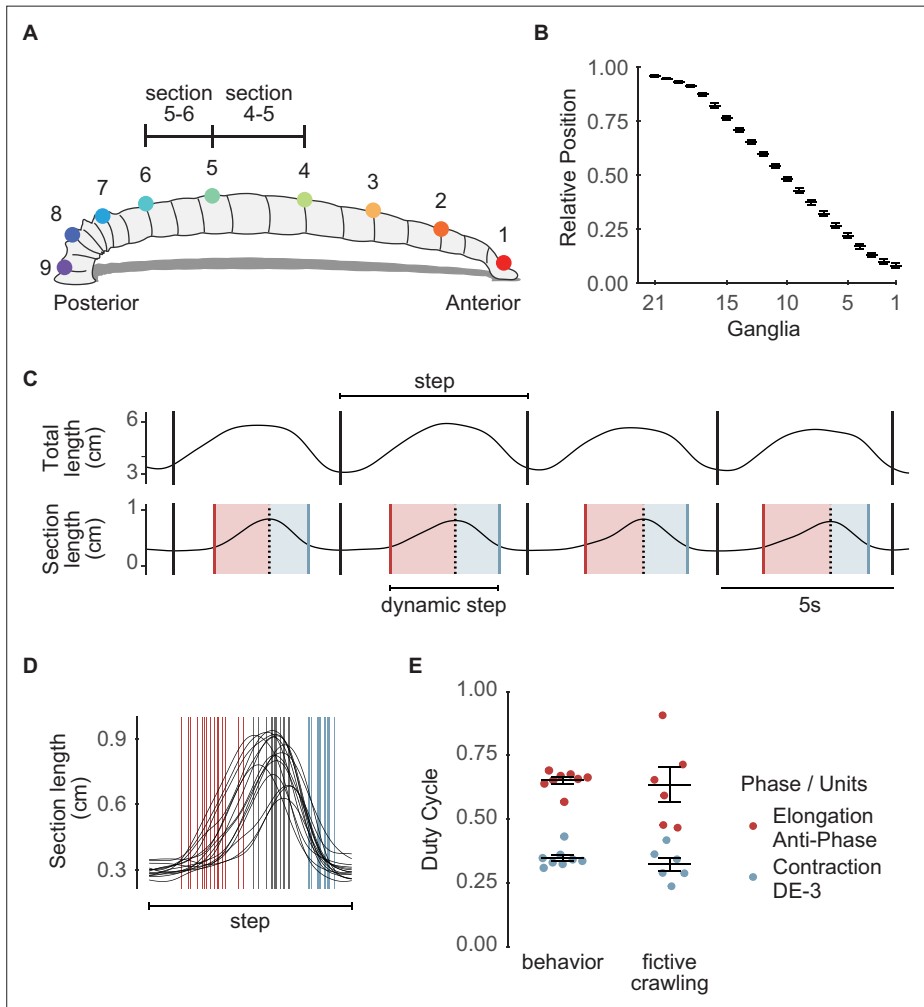

**Figure 5.** Analysis of leech crawling. (**A**) Schematic representation of a leech with nine points as painted on their dorsal midline. The sections between points 4–5 and 5–6 comprise ganglia 8–15 that were used for the electrophysiological analysis. (**B**) Relative position of midbody ganglia 1–21. Mean and standard error of the mean are shown (n=3) (**Kearney et al., 2022**). (**C**) Representative example of the changes in total length and sections 4–5 of a leech during four steps as the animal was crawling. The limits of each step are indicated by black vertical lines (see 'Materials and methods'); red and blue vertical lines mark the start of elongation and the end of contraction, respectively; black dashed vertical lines indicate the maximal section length. Light red and blue shades indicate elongation and contraction phases, respectively. (**D**) Superimposed length curves of a section across random multiple steps (n=15 of 76 total steps for this animal) of one leech. (**E**) Duty cycle of the elongation and contraction phases measured in crawling behavior or in fictive crawling. Behavioral variables were calculated as the duration of each phase relative to the sum of elongation and contraction phases (dynamic step in **C**), excluding isometric phases. Points represent the mean for each leech, while bars and lines indicate the mean ± SEM across all leeches (n=249 steps from eight leeches). For the duty cycle of DE-3 and anti-phase units, points correspond to the mean for each ganglion, with bars and lines showing the mean ± SEM across all ganglia (n=4 cycles for each DE-3 or anti-phase motoneuron, six ganglia, six leeches).

## Comparison of the motor pattern in single ganglia and in intact animals

The assumption that the crawling motor pattern described on the basis of extracellular recordings in isolated single ganglia reflects key characteristics of the behavioral pattern requires qualitative and quantitative validation. To this end, intact leeches were monitored during crawling. The analysis was performed by tracking nine dots painted along the dorsal longitudinal midline of the leech, which divided the body into eight sections (*Figure 5A*). Because ganglia are located in highly conserved positions relative to the length of the animal (*Figure 5B*, *Kearney et al., 2022*), the relative distance from the head can be used to infer which ganglia are included within the eight sections mentioned above. We focused the analysis on sections delimited by points 4–5 and 5–6, that correspond to the range of segmental ganglia included in the ex vivo studies (ganglia 8–15) (*Figure 5A and B*). For electrophysiological data, the frame of analysis was the cycle of DE-3 rhythmic activity and, equivalently, for the behavioral data, the frame of analysis was the crawling step (see 'Materials and methods').

*Figure 5C* shows a representative trace of a series of whole length waves that were divided into discrete steps (see 'Materials and methods'), and the corresponding measurements in the section delimited by the dots 4 and 5. Within each crawling step, four stages were identified in the studied sections (*Figure 5C*): the elongation stage (marked by an increase in section length), the contraction stage (marked by a decrease in section length), and two isometric stages flanking the dynamic stages. The plot in *Figure 5D* shows the section length of different steps by the same animal, normalized by the step duration, and illustrates the variability of the step dynamics.

For a quantitative comparison of the in vivo and ex vivo experiments, the following variables were evaluated: period of the rhythmic pattern and duty cycle of the different phases. While the average cycle period (end-to-end) recorded in isolated ganglia was 15.22±3.21 s (n=6), the average duration of the leech step was 7.49±0.50 s (n=6). These values are similar to those reported in previous studies (*Puhl and Mesce, 2008*, for single ganglia; and *Stern-Tomlinson et al., 1986*; *Baader and Kristan, 1992* and *Puhl and Mesce, 2008*, for intact animals). Thus, in the intact animal, the rhythmic activity was twice as fast as in the isolated ganglion, suggesting that peripheral signals may accelerate the rhythmic behavior, as already described in other animals (*Pulver et al., 2015*; *Fushiki et al., 2016*). However, we cannot rule out that activation of the rhythmogenic circuit in isolated ganglia may fail to activate a circuit component responsible for speeding the rhythm.

It seems natural to correlate the activity of the anti-phase and in-phase units of ex vivo studies with the elongation and contraction stages of in vivo behavior, respectively. But the isometric stages displayed in vivo at the section level have no obvious counterpart in the electrophysiological recordings of motoneurons in the isolated ganglia. It is important to consider that the rhythmic movement of successive segments along the antero-posterior axis of the animal requires a delay signal that allows the appropriate propagation of the metachronal wave (*Puhl et al., 2012*; *Puhl and Mesce, 2010*), and this signal is probably absent in the isolated ganglion. In consequence, for the comparison of the duty cycle, the in vivo isometric stages were ignored, and only the duration of the dynamic fraction of the step was taken into consideration. In turn, the behavioral duty cycle was calculated as the ratio between the elongation or contraction stage duration over the dynamic step duration. Thus measured, the duty cycle of the elongation stage was 0.65±0.01 and that of the contraction was 0.35±0.01 (*Figure 5E*). These values are noticeably close to the duty cycle of the anti-phase units (0.64±0.07) and of the DE-3 motoneurons (0.32±0.03) (*Figure 5E*) measured in the isolated ganglia.

These results show that the dynamic pattern described by multiple motoneurons recorded in isolated ganglia reflects the phase relation observed in the corresponding body section in vivo.

## Effect of NS on the activity of the different types of motoneurons during *crawling*

The results shown in *Figure 5* support the view that the activity recorded in single isolated ganglia reflects the neuronal control of the rhythmic motor pattern underlying crawling. Based on this premise, the recording of multiple units presented above was used to evaluate the influence of NS on units active at different *crawling* stages (*Figure 6Ai–ii*). Previous work (*Rodriguez et al., 2012*) and the experiments presented in *Figure 3* suggest that NS depolarization releases motoneuron DE-3 from the inhibitory influence of the premotor neuron. The suggested interpretation postulates that NS depolarization, by turning off the rectifying junctions, prevents the transmission of NS inhibitory inputs onto DE-3. Given the widespread connectivity of NS with all the excitatory motoneurons, it is

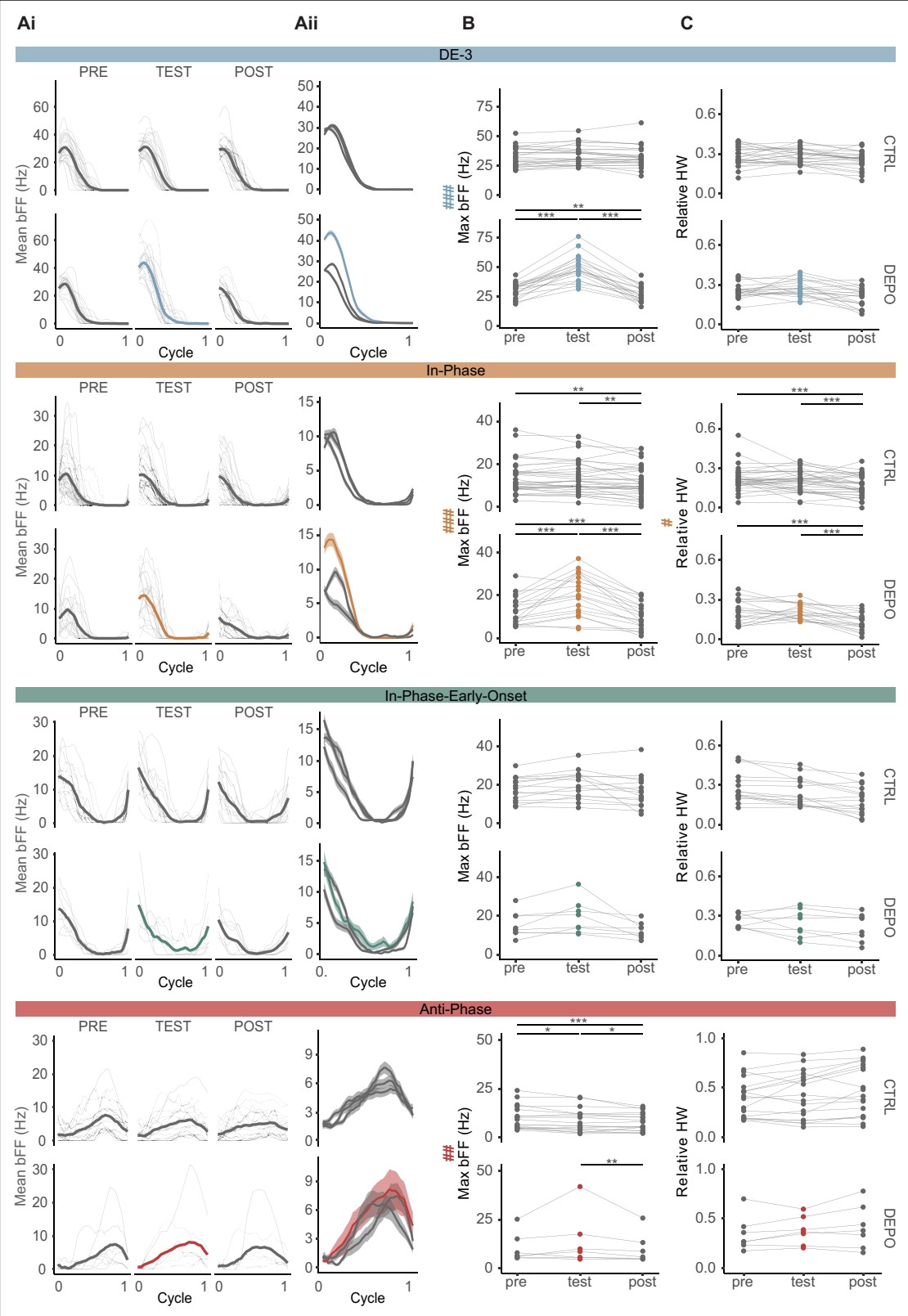

**Figure 6.** Effect of nonspiking (NS) on the different phases of *crawling*. (**Ai**) Mean binned firing frequency (bFF) across cycles in pre, test, and post epochs for each unit within the four groups described in *Figure 4*. For each unit type, the upper line of graphs presents control experiments, where no current was injected in NS, and the bottom line, depolarization experiments, where 5–7 nA current steps were injected in NS during the test epoch. Thin lines show the mean bFF (including 4–6 cycles) for each experiment, and thick lines show the overall mean. (**Aii**) Superimposition of the overall bFF

*Figure 6 continued on next page*

*Figure 6 continued*

mean of the three epochs. The shaded area presents the standard error of the mean. (**B, C**) Maximum bFF (Max bFF) and relative half width (relative HW) measured in each individual unit. (**A–C**) are plotted for each of the four groups of units indicated. Lines connect the mean values across each epoch within the same experiment. Mixed-effects models with two fixed factors were performed; on the left of the (**B**) and (**C**) panel, # indicates a significant interaction between treatment and epoch factors; * indicates significant differences for pairwise simple contrasts between epochs, within each treatment. *, **, and *** (or #) indicate p<0.05, p<0.01, and p<0.001, respectively. **Supplementary file 2** describes the statistical models performed. For DE-3, n=26 control and 20 depolarization ganglia, from 46 ganglia; for in-phase, n=38 units for control and 21 units for depolarization, from 30 ganglia; for anti-phase, n=18 units for control and 7 units for depolarization experiments, from 22 ganglia; and for in-phase-early-onset, n=15 units for control and 8 units for depolarization experiments, from 23 ganglia.

expected that NS manipulation affects the activity of all the units that fire in phase with DE-3 when NS receives inhibitory inputs (**Figure 2A**). In contrast, units active in anti-phase with DE-3 should not be affected as NS does not receive inhibitory inputs during that stage.

To evaluate the effect of NS depolarization, the activity of the motoneurons was characterized using two metrics: maximum bFF (Max bFF) to assess activity level, and relative half width (Rel HW) to evaluate the duty cycle. To enable a comparison with the other units, the data for DE-3 was reanalyzed in this manner.

NS depolarization produced an increase in DE-3 Max bFF of around 60% (as shown above), while this variable remained stable throughout control experiments (**Figure 6B**). Different from DE-3, in-phase units showed a marked decrease in the maximum bFF across time in control conditions, but similar to DE-3, this variable underwent an increase of around 50% upon NS depolarization (**Figure 6B**). Given the temporal drop detected in the control series, this increase is probably an underestimation of the NS effect on the in-phase firing frequency.

The anti-phase units also showed a slight but significant decrease in the maximum bFF across time in control conditions, but in this case, the effect of NS depolarization was limited to counteracting this decrease (**Figure 6B**). While in control conditions, anti-phase motor units exhibited a decrease of around 15% between pre and test epochs, the depo experiments showed no decrement between these two epochs. Notwithstanding, control and depo experiments showed a marked decrease in bFF from test to post (15% and a 20% decrease, respectively). NS depolarization did not affect the maximum bFF of the in-phase-early-onset units (**Figure 6B**).

Regarding the relative HW, the effect of NS depolarization was limited to in-phase units. Our results show that the relative HW of these units tended to fall across epochs in control conditions and that releasing NS depolarization enhanced this tendency (**Figure 6C**). In control experiments, the relative HW decreased by about 15% between test and post epochs, and in the depo experiments, the decrease was of around 40%.

The statistical analyses, as well as the estimated ratios between epochs for each case, are detailed in **Supplementary file 2**.

Taken together, the results support the hypothesis presented earlier: the inhibitory signals received by NS in phase with the contraction stage modulated the activity of the motoneurons active in this instance. It is worth noting that the fact that the firing frequency of anti-phase MNs was not affected by NS depolarization indicates that this experimental manipulation did not cause a generalized change in the activity of MNs.

## Discussion

The present study shows that phase-specific inhibitory signals, delivered by the functional analogue of Renshaw cells in the leech (**Szczupak, 2014**), modulate the degree of excitation of the motoneurons responsible for the contraction stage of *crawling*. This effect was phase-specific, as motoneurons active in the rest of the cycle were not affected. NS is a premotor neuron that operates at the center of a recurrent inhibitory circuit (**Rela and Szczupak, 2003**; **Rodriguez et al., 2009**) and thus the initial hypothesis suggested that the effect on *crawling* was mediated entirely by this circuit. However, the results show that the inhibitory signal did not originate at the level of the motoneurons themselves but, most probably, at the crawling oscillator. The marked temporal correlation between the neurons active at the contraction stage and NS hyperpolarizations suggests that the excitatory signals onto the motoneurons and the hyperpolarizing signals onto NS were driven by the same upstream element

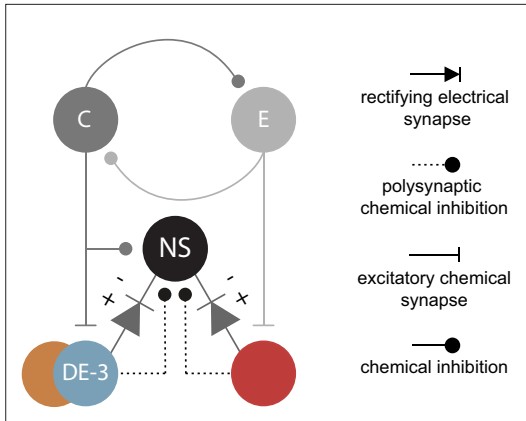

**Figure 7.** Nonspiking (NS) in the *crawling* circuitry. Putative neuronal circuitry as described in *Figure 1C*, including the connection between the oscillatory circuit and NS. The colors used here match those identifying the units in *Figures 4 and 6*.

rectifying electrical synapse

polysynaptic chemical inhibition

excitatory chemical synapse

chemical inhibition

(e.g., the element in the oscillator that controls the contraction stage). This result further feeds the analogy between vertebrate Renshaw cells and leech NS neurons in that the formers are not only targeted by motoneuron inputs but by rhythmogenic premotor elements as well (*Nishimaru et al., 2006*). Nonetheless, the results do not rule out that the existing recurrent inhibitory circuit contributes to the hyperpolarization of NS.

On the other hand, the fact that in-phase-early-onset activity was also unaffected by NS depolarization confirms that these motoneurons constitute a distinct group from the in-phase units. Furthermore, this suggests that these units might be inhibitory motoneurons that are not connected to the NS via the recurrent inhibitory circuit (*Rela and Szczupak, 2003*; *Wadepuhl, 1989*).

In summary, we propose that the crawling oscillator releases a dual signal onto the motoneurons that control the contraction stage: a direct excitatory input that elicits the rhythmic bursts and a concurrent inhibitory input via NS (*Figure 7*). Consistent with this interpretation, the activity level of the anti-phase units was not influenced by these inhibitory signals.

The corroboration of a dual counteracting set of actions on motoneuron output calls for a functional interpretation. In a characterization of the electrophysiological properties of leech motoneurons, it was shown that DE-3 can reach a firing frequency that exceeds 100 Hz (*Bernardo Perez-Etchegoyen et al., 2012*) while the maximal muscle tension that this motoneuron can evoke is achieved at around 25 Hz (*Mason and Kristan, 1982*). Additionally, it is relevant to consider that the population of motoneurons that activate longitudinal muscles is linked by ohmic coupling that can result in mutual excitation (*Fan et al., 2005*; *Ort et al., 1974*; *Ashaber et al., 2021*). As a result, when all the activators of longitudinal muscles are excited during the contraction stage, the excitatory spread can lead to runaway motoneuron activity (*Hennequin et al., 2017*). An inhibitory signal imposed on NS can spread to all the motoneurons simultaneously (*Rela and Szczupak, 2003*; *Rodriguez et al., 2009*; *Wadepuhl, 1989*) and thus limit the activity of the whole motoneuron population.

Previous work showed that leech motoneurons are subjected to a generalized inhibition mediated by GABA (*Baca et al., 2008*) while the electrically mediated inhibition exerted by NS shown here is phase-specific. The existence of concurrent excitation and inhibition has been revealed in vertebrate motor networks, where it was proposed as a balancing factor (*Kishore et al., 2014*; *Parkis et al., 1999*; *Petersen et al., 2014*). In line with the present results, experiments performed in zebrafish showed that in-phase inhibition originates from a set of premotor neurons that comprise the Renshaw cells (*Callahan et al., 2019*). Neurophysiological studies in the leech offer the particular advantage of allowing electrical manipulation of individual neurons in adults, where they can be readily recognized by their anatomical and electrophysiological characteristics. In this sense, the present study offers clear-cut evidence of the dual signal motoneurons receive in the course of motor control from an identified premotor element.

The results further highlight the notion that knowing the connectivity among neurons is not sufficient to deduce its functional operation, as the dynamical interplay among them is highly relevant to understand its physiological role.

## Structure of *crawling* based on MN activity and behavior

Evaluation of phase specificity required a more comprehensive tracking of motoneuron activity throughout the cycle, and we addressed this demand through the analysis of the activity of multiple motoneurons. In the present study, we show that in the course of *crawling*, DE-3 neurons fire concurrently with neurons that reach the periphery mainly through the PP and the AA roots. Based on the

classical description of *Ort et al., 1974*, the spikes in phase with DE-3 recorded in the PP root are likely to correspond to VE-4 (ventral-excitor 4), DE-5, DE-7, and VE-8, while those recorded in AA are likely to correspond to DE-107 and VE-108. Intracellular recordings of VE-4 and VE-5 showed indeed the coincident firing with DE-3 (*Baader, 1997*; *Eisenhart et al., 2000* and *Puhl and Mesce, 2008*).

To learn how the motoneuron readout compares with the rhythmic body movements that take place in crawling, we presented behavioral studies that focused on the sections of the leech body that match the range of ganglia used in the electrophysiological studies. The comparison shows two main differences. The period of the rhythmic animal behavior was half that measured in the fictive behavior at the isolated ganglia. In addition, at the behavioral level, the actual elongation-contraction gestures were flanked by isometric stages, while in dopamine-induced *crawling* studied in isolated ganglia, the rhythmic activity of the motoneurons appears as the alternation between units in-phase with DE-3 and anti-phase of DE-3, with no clear references for isometric phases. The absence of known descending brain signals (*Puhl et al., 2012*) and/or peripheral signals is assumed as important factors in determining the cycle period and the sequence at which the different behavioral stages take place. However, it is noteworthy that taking the isometric stages aside, the sequence of events, and the proportion of the active cycle dedicated to elongation and contraction were remarkably similar in both experimental settings. This suggests that the network activated in the isolated ganglion is the one underlying the dynamic aspects of the motor behavior at any given section of the animal.

## Materials and methods

### Biological preparation

Leeches (*Hirudo* sp.), weighing 1–3 g, were sourced from commercial suppliers (Niagara Leeches, Cheyenne, WY, USA, and Biopharm, UK) and kept at 20°C in artificial pond water. These animals are hermaphrodites.

Studies were conducted in isolated single ganglia obtained from segments 8 to 13. The ganglia were dissected with one dorsal posterior (DP), posterior (PP), and anterior (AA) nerve roots attached. The tissue was bathed in normal saline (in mM: 115 NaCl, 4 KCl, 1.8 CaCl$_2$, 1 MgSO$_4$, 10 HEPES, 10 glucose; pH 7.4) at room temperature (20–25°C) and pinned to Sylgard (Dow Corning) in a recording chamber. The sheath covering the ganglion was dissected away, exposing the neuronal cell bodies to the external solution.

### Electrophysiological recordings

Intracellular somatic recordings were conducted using microelectrodes pulled from borosilicate capillary tubing (A-M Systems, Inc, Carlsborg, WA, USA), filled with 3 M potassium acetate (resistance 20–40 MΩ). The electrodes were connected to an Axoclamp 2B amplifier (Axon Instruments; Union City, CA, USA) operating in bridge mode. Extracellular activity from DP, PP, and AA nerves was recorded using suction electrodes connected to a differential a.c. amplifier (Neuroprobe 1700, A-M Systems, Inc). Intra- and extracellular recordings were digitized through an analog-digital converter (Digidata 1440, Axon Instruments) and acquired with Clampex 9.2 (Axon Instruments) at a sampling rate of 20 kHz. Premotor NS neurons were identified by their soma location and electrophysiological properties (*Burgin and Szczupak, 2003*; *Rela et al., 2009*).

To induce fictive crawling (*crawling*), ganglia were superfused with 75 μM dopamine hydrochloride (Sigma-Aldrich, St. Louis, MO, USA), freshly prepared at the beginning of each experimental day (*Puhl and Mesce, 2008*). Only one episode was evoked per ganglion. The rhythmic motor pattern was monitored via extracellular recording of the DP nerve, where the largest spike corresponds to the DE-3 motoneuron, one of the motoneurons innervating the longitudinal muscles (*Ort et al., 1974*). In the present study, the minimum and maximum mean period values (end-to-end) in isolated ganglia were 6.32 and 31.24 s, respectively.

When testing the effect of NS neurons, depolarizing pulses of +5 nA to +7 nA were injected into its soma through the intracellular electrode. Between four and five *crawling* cycles before and after a depolarizing pulse were left unaffected to obtain control pre- and post-rhythmic epochs.

## Behavioral experiments

Leeches were anesthetized by placing individuals at −20°C for 5–10 min. Then the skin of the animal was dried, and nine points of water-based paint were applied along its dorsal longitudinal midline (*Figure 5A*). When the animals recovered, they were allowed to crawl onto a graph paper sheet covered with a transparent plastic surface. Crawling was filmed at 60 fps with a digital camera (Sony ILCE-A6400) supported above the arena. The position of each point painted on the leech skin was tracked using DeepLabCut version 2.3.5 (*He et al., 2016*; *Insafutdinov et al., 2016*; *Mathis et al., 2018*; *Nath et al., 2019*). To this end, 338 frames randomly selected from 66 videos of 8 leeches were used for manually labeling the painted dots and used for training a ResNet_50-based neural network for automating the detection.

To characterize crawling, steps were defined by the frames in which the algorithmic sum of the following values was at a minimum: (1) the whole length of the leech, (2) the velocity of the most anterior dot, and (3) the velocity of the most posterior dot (*Figure 5C*). Steps where DeepLabCut failed to detect any of the nine points on the animal were discarded. After dividing the animal displacement into steps, we performed a quantitative analysis of the changes in the length of the animal sections (defined between each pair of points) over time. Specifically, the sections delimited by the painted points 4–5 and 5–6 were chosen since they include the range of ganglia considered in the electrophysiological studies (8–15). The length traces of each leech were filtered using a Gaussian filter (sigma of 0.25 s). For each step, the duration of the elongation and contraction stages within a section was calculated by using the maximum length as the boundary between these phases. The onset of elongation and the end of contraction were determined by detecting knee points, as described by *Satopaa et al., 2011*, *Figure 5C*.

## Spike sorting

Extracellular recordings were spike sorted using a custom-written algorithm which relies on a template matching-based method based on *Pouzat et al., 2002*, incorporating a t-SNE model (*Pedregosa et al., 2011*; *Maaten and Hinton, 2008*). Because spike sorting is sensitive to misclassification, the resulting output was manually curated to address whether a single cluster of waveforms was self-consistent with a single neuron (*Hill et al., 2011*). During manual curation, all clusters of putative spike events were evaluated. Clusters containing two well-defined groups of waveforms were split, and pairs of clusters with similar waveforms and temporally coordinated activity were merged. Clusters that had inconsistent waveforms and could not be split into clusters with well-defined waveforms were discarded. After this procedure, each cluster was considered a single motor unit.

To select for rhythmic units, autocorrelations were performed considering the first four cycles of the fictive behavior. Units whose autocorrelogram included at least 2 peaks with a correlation coefficient >0.25 were considered rhythmic.

## Data processing

DE-3 spike times were derived from DP nerve recordings using the spike sorting algorithm mentioned above. DE-3 bursts were considered as such if they comprised at least 10 spikes with a mFF of 8 Hz, where the separation between two subsequent spikes occurred with a maximum separation of 0.5 s. Burst duration was calculated as the time difference between the last and first DE-3 spikes in each burst. The cycle period was calculated in two different ways: as the time difference between the last (end-to-end), or the first spikes (start-to-start) of subsequent DE-3 bursts, and is reported accordingly. The duty cycle was calculated as the ratio between burst duration and period. The mFF of each burst was calculated as the number of spikes over the burst duration. The bFF of DE-3, or any other unit, was obtained by binning the number of spikes across the recording using a 0.05 s or a 0.5 s interval, for *Figures 2 and 6*, respectively. For cross-correlation analysis, the DE-3 bFF was filtered using a Gaussian filter (sigma of 150 ms).

For the cross-correlation between NS membrane potential and DE-3 firing frequency, the intracellular recording of NS was resampled to match the data rate of DE-3 bFF. To determine the amplitude of NS hyperpolarization, a baseline membrane potential was established using a Gaussian filter (sigma of 5 s) applied to the entire trace. The amplitude was calculated as the difference between the baseline value at the start of the DE-3 burst and the minimum membrane potential reached by NS during the corresponding burst (see *Figure 2A*).

When NS depolarization was implemented, the cycles including the onset or offset of the current pulse were excluded, and the traces were divided into three epochs: pre, test, and post. Control experiments were treated similarly.

When analyzing the activity of the units obtained by spike sorting, the maximum bFF value and relative half width were obtained from curves displaying bFF versus cycle phase (time normalized to the period of each cycle). The relative half-width was calculated as the width at half the maximum bFF value.

Data processing was performed using custom-written Python scripts.

## Statistical analysis

All statistical analyses were performed using R (*R Development Core Team, 2024*). The goodness of fit for each model was analyzed using the DHARMa package. For every analysis, the structure of random factors was kept maximal whenever possible. The specific random factors considered in each case are detailed in *Supplementary files 1 and 2*.

To analyze the correlation between the amplitude of NS hyperpolarization and different *crawling* features (*Figure 2D–G*), the lmer function from the lmerTest package (*Kuznetsova et al., 2017*) was used to perform linear mixed models (LMM) with the experiment as a random factor.

To analyze the effect of NS depolarization on motoneuron activity, the data was analyzed using mixed-effects models with two fixed factors: epoch (pre, test, post) and treatment (control, depolarization). All models included motor units as a random factor. Each ganglion was also included as a random factor. The effect of NS depolarization was analyzed with the glmmTMB function from the glmmTMB package (*Brooks et al., 2017*) to perform a generalized linear mixed model (GLMM) or with the lmer function to perform a LMM. For the GLMM, we used a Student $t$ distribution (t_family) with the identity link or Gamma family with a log link as indicated in *Supplementary files 1 and 2*. The significance of a factor was determined with a likelihood-ratio test for the GLMM or an F-test for the LMM. This was done by comparing the model including the factor with a model dropping that factor. Pairwise simple contrasts were performed within the treatment factor and between the epoch factor using the *emmeans* package, and the Tukey correction was applied to obtain the p values.

To analyze a possible temporal drift on *crawling* features, we considered the data of control experiments and used mixed-effects models with epoch (pre, test, post) as a fixed factor and units as a random factor. *Supplementary files 1 and 2* specify the models used for each analysis.

## Acknowledgements

We thank Drs. Violeta Medan, Graciela Kearney, Alejandro Cámera, and Ronald Calabrese for highly valuable comments on the manuscript. We also thank Dr. Julieta Laurino for the highly valuable discussions on statistics. This work was supported by the following grants to LS: PICT 2016-2073 from Agencia Nacional de Promoción Científica y Tecnológica; UBACyT 20020150100179BA from Universidad of Buenos Aires and Human Frontier Science Program (RGP0060/2019).

## Additional information

### Funding

| Funder | Grant reference number | Author |
| --- | --- | --- |
| Human Frontier Science Program | RGP0060/2019 | Lidia Szczupak |
| Universidad de Buenos Aires | 20020220100096BA | Lidia Szczupak |
| Agencia Nacional de Promoción de la Investigación, el Desarrollo Tecnológico y la Innovación | PICT-2021-I-A-00080 | Lidia Szczupak |

| Funder | Grant reference number | Author |
|--------|------------------------|--------|

The funders had no role in study design, data collection and interpretation, or the decision to submit the work for publication.

## Author contributions

Martina Radice, Conceptualization, Data curation, Software, Formal analysis, Investigation, Visualization, Methodology, Writing – original draft, Writing – review and editing; Agustin Sanchez Merlinsky, Software, Investigation, Methodology; Federico Yulita, Formal analysis, Investigation, Methodology; Lidia Szczupak, Conceptualization, Resources, Data curation, Supervision, Funding acquisition, Validation, Investigation, Visualization, Writing – original draft, Project administration, Writing – review and editing

## Author ORCIDs

Martina Radice ⓘ https://orcid.org/0009-0009-5455-2998
Lidia Szczupak ⓘ https://orcid.org/0000-0001-5467-0059

## Ethics

not applicable.

Reviewer #1 (Public review): https://doi.org/10.7554/eLife.104921.4.sa1
Reviewer #2 (Public review): https://doi.org/10.7554/eLife.104921.4.sa2
Author response https://doi.org/10.7554/eLife.104921.4.sa3

# Additional files

## Supplementary files

Supplementary file 1. Statistical models performed for the effect analysis of NS depolarization on *crawling* features and DE-3 activity during the fictive behavior. For each analyzed variable, the use of either a linear mixed model (LMM) or a generalized linear mixed model (GLMM) is specified, together with the structure of the random effects. In the case of GLMMs, the distribution and the corresponding link function are also specified. For each model, significance levels, p values, and ratios or estimated values are reported.

Supplementary file 2. Statistical models performed for the effect analysis of NS depolarization on different groups of units activity during *crawling*. For each analyzed variable, the use of either a linear mixed model (LMM) or a generalized linear mixed model (GLMM) is specified, together with the structure of the random effects. In the case of GLMMs, the distribution and the corresponding link function are also specified. For each model, significance levels, p-values, and ratios or estimated values are reported.

MDAR checklist

## Data availability

The data and codes are available in https://doi.org/10.5061/dryad.ksn02v7jv.

The following dataset was generated:

| Author(s) | Year | Dataset title | Dataset URL | Database and Identifier |
|-----------|------|---------------|-------------|-------------------------|
| Radice M, Szczupak L | 2025 | Phase-specific premotor inhibition modulates leech rhythmic motor output | https://doi.org/10.5061/dryad.ksn02v7jv | Dryad Digital Repository, 10.5061/dryad.ksn02v7jv |

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
