## [Editor Report · eLife Assessment]

The medicinal leech preparation is an amenable system in which to understand the neural basis of locomotion. Here, a previously identified non-spiking neuron was studied in leech and found to alter the mean firing frequency of a crawl-related motoneuron, which fires during the contraction phase of crawling. The findings are **valuable**, and the experiments were diligently done and considered **solid**. The results lay a foundation for additional studies in this system.

---

## [Referee Report · Reviewer #1 (Public review)]

The medicinal leech preparation is an amenable system in which to understand how the underlying cellular networks for locomotion function. A previously identified non-spiking neuron (NS) was studied and found to alter the mean firing frequency of a crawl-related motoneuron (DE-3), which fires during the contraction phase of crawling. The data are solid. Identifying upstream neurons responsible for crawl motor patterning is essential for understanding how rhythmic behavior is controlled.

---

## [Referee Report · Reviewer #2 (Public review)]

This study by Radice et al., takes advantage of the very well-established leach preparation to investigate questions related to motor control, more precisely the question of how the activity of motoneurons taking part in leach crawling behavior are finely tuned.

The paper is overall well written. The findings are clearly presented, and the data seems solid overall.

---

## [Author Response]

The following is the authors’ response to the previous reviews

**Reviewer #1 (Public review):**
The medicinal leech preparation is an amenable system in which to understand how the underlying cellular networks for locomotion function. A previously identified non-spiking neuron (NS) was studied and found to alter the mean firing frequency of a crawl-related motoneuron (DE-3), which fires during the contraction phase of crawling. The data are mostly solid. Identifying upstream neurons responsible for crawl motor patterning is essential for understanding how rhythmic behavior is controlled.Review of Revision:On a positive note, the rationale for the study is clearer to me now after reading the authors' responses to both reviewers, but that information, as described in the authors' responses, is minimally incorporated into the current revised paper. Incorporating a discussion of previous work on the NS cell has, indeed, improved the paper.I suggested earlier that the paper be edited for clarity but not much text has been changed since the first draft. I will provide an example of the types of sentences that are confusing. The title of the paper is: "Phase-specific premotor inhibition modulates leech rhythmic motor output". Are the authors referring to the inhibition created by premotor neurons (e.g., on to the motoneurons) or the inhibition that the premotor neurons receive?

In this case, this is an interesting ambiguity: NS is inhibited and that inhibition is directly transmitted to the motoneurons because both cells are electrically coupled. We believe that the title does not disguise the findings conveyed by the manuscript.

I also find the paper still confusing with regard to the suggested "functional homology" with the vertebrate Renshaw cells. When the authors set up this expectation of homology (should be analogy) in the introduction and other sections of the paper, one would assume that the NS cell would be directly receiving excitation from a motoneuron (like DE-3) and, in turn, the motoneuron would then receive some sort of inhibitory input to regulate its firing frequency. Essentially, I have always viewed the Renshaw cells as nature's clever way to monitor the ongoing activity of a motoneuron while also providing recurrent feedback or "recurrent inhibition" to modify that cell's excitatory state. The authors present their initial idea below on line 62. Authors write: "These neurons are present as bilateral pairs in each segmental ganglion and are functional homologs of the mammalian Renshaw cells (Szczupak, 2014). These spinal cord cells receive excitatory inputs from motoneurons and, in turn, transmit inhibitory signals to the motoneurons (Alvarez and Fyffe, 2007)."

We agree with Reviewer #2: the correct term is "analogous," not "homologous." Thanks for pointing this out. We changed the term throughout the text.

The Reviewer is also right in the appreciation of the role of Renshaw cells. NS plays exactly the role that the Reviewer expresses. The ONLY difference is that NS is inhibited by the motoneurons, and in turn transmits this inhibition to the motoneurons via the rectifying electrical junctions. Attending the confusion that our description caused in the Reviewer, we have modified the cited sentence accordingly now in lines 65-67.

Minor note:I suggest re-writing this last sentence as "these" is confusing. Change to: 'In the spinal cord, Renshaw interneurons receive excitatory inputs from motoneurons and, in turn, transmit inhibitory signals to them (Alvarez and Fyffe, 2007).'

Please, see the changes mentioned above.

Furthermore, the authors note that (line 69 on): "In the context of this circuit the activity of excitatory motoneurons evokes chemically mediated inhibitory synaptic potentials in NS. Additionally, the NS neurons are electrically coupled......In physiological conditions this coupling favors the transmission of inhibitory signals from NS to motoneurons." Based on what is being conveyed here, I see a disconnect with the "functional homology" being presented earlier. I may be missing something, but the Renshaw analogy seems to be quite different compared to what looks like reciprocal inhibition in the leech. If the authors want to make the analogy to Renshaw cells clearer, then they should make a simple ball and stick diagram of the leech system and visually compare it to the Renshaw/motoneuron circuit with regard to functionality. This simple addition would help many readers.

We have simplified the description regarding the Renshaw cell (lines 65-67) to avoid the “details” of the connectivity between the two circuits.

This report focuses on NS neurons and their role in crawling; we mention the analogy with Renshaw cells to widen the interest of the results. We do not think that making a special diagram to compare how the two neurons play a similar role via different connections among the players is useful in the context of this manuscript.

The Abstract, Authors write (line 19), "Specifically, we analyzed how electrophysiological manipulation of a premotor nonspiking (NS) neuron, that forms a recurrent inhibitory circuit (homologous to vertebrate Renshaw cells)...."First, a circuit would not be homologous to a cell, and the term homology implies a strict developmental/evolutionary commonality. At best, I would use the term functionally analogous but even then I am still not sure that they are functionally that similar (see comments above).

Reviewer #2 is right. We changed the sentence in line 20.

Line 22: "The study included a quantitative analysis of motor units active throughout the fictive crawling cycle that shows that the rhythmic motor output in isolated ganglia mirrors the phase relationships observed in vivo." This sentence must be revised to indicate that not all of the extracellular units were demonstrated to be motor units. Revise to: "The study included a quantitative analysis of identified and putative motor units active throughout the fictive crawling cycle that shows.....'Line 187 regarding identifying units as motoneurons: Authors write, "While multiple extracellular recordings have been performed previously (Eisenhart et al., 2000), these results (Figure 4) present the first quantitative analysis of motor units activated throughout the crawling cycle in this type of recordings." The authors cannot assume that the units in the recorded nerves belong only to motoneurons. Based on their first rebuttal, the authors seem to be reluctant to accept the idea that the extracellularly recorded units might represent a different class of neurons. They admit that some sensory neurons (with somata located centrally) do, indeed, travel out the same nerves recorded, but go on to explain why they would not be active.The leech has a variety of sensory organs that are located in the periphery, and some of these sensory neurons do show rhythmic activity correlated with locomotor activity (see Blackshaw's early work). The numerous stretch receptors, in fact, have very large axons that pass through all the nerves recorded in the current paper.In Fig. 4, it is interesting that the waveforms of all the units recorded in the PP nerve exhibit a reversal in waveform as compared to those in the DP nerve, which might indicate (based on bipolar differential recording) that the units in the PP nerve are being propagated in the opposite direction (i.e., are perhaps afferent). Rhythmic presynaptic inhibition and excitation is commonly seen for stretch receptors within the CNS (see the work of Burrows) and many such cells are under modulatory control.Most likely, the majority of the units are from motoneurons, but we do not really know at this point. The authors should reframe their statements throughout the paper as: 'While multiple extracellular recordings have been performed previously (Eisenhart et al., 2000), these results (Figure 4) present the first quantitative analysis of multiple extracellular units, using spike sorting methods, which are activated throughout the crawling cycle.' In cases where the identity of the unit is known, then it is fine to state that, but when the identity of the unit is not known, then there should be some qualification and stated as 'putative motor units'

We understand the concern of Reviewer #2 regarding the type of neurons active during dopamine-induced crawling in isolated ganglia. However, we believe there is sufficient evidence to support that the recorded spikes originate from motoneurons. As readers may share the same concern, we have added a paragraph explaining why spikes from somatic sensory neurons such as P or T cells, or from stretch receptors, are unlikely to contribute (lines 206-214). We included the term putative in the abstract.

The Methods section:Needs to include the full parameters that were used to assess whether bursting activity was qualified in ways to be considered crawling activity or not. Typically, crawl-like burst periods of no more than 25 seconds have been the limit for their qualification as crawling activity. In Fig 2F, for example, the inter-burst period is over 35 seconds; that coupled with an average 5 second burst duration would bring the burst period to 40 seconds, which is substantially out of range for there to be bursting relevant to crawl activity. Simply put, long DE-3 burst periods are often observed but may not be indicative of a crawl state as the CV motoneurons are no longer out of phase with DE-3. A number of papers have adopted this criterion.

We now indicate in the methods the range of period values measured in our experiments. For the reviewer informatio we show here histograms depicting the variability of period and duty cycle values recorded in our experiments (control conditions). The Reviewer can see that the bursting activity of DE-3 fall within what has been published.

**Author response image 1. sa3fig1:** Crawling in isolated ganglia. A. Histogram of periods end-to-end during crawling in isolated ganglia. The dotted line indicates the mean obtained from the averages of all experiments. The solid black line represents the mean of all cycles across all experiments. B. As in A, for the duty cycle calculated using end-to-end periods. (n = 210 cycles from 45 ganglia obtained from 32 leeches in all cases).

**Reviewer #1 (Recommendations for the authors):**
Minor comments-Line 100: "In the frame of the recurrent inhibitory circuit, NS is the target of inhibitory signals". Suggestion: 'Within the framework of the recurrent inhibitory circuit, NS is the target of inhibitory signals.'

Changed as suggested (line 107).

Line 163: "This series of experiments proves that, as predicted based on the known circuit (Figure 164 1C), inhibitory signals onto NS premotor neurons were transmitted to DE-3 motoneurons and counteracted their excitatory drive during crawling, limiting their firing frequency". I think this sentence is too strong plus needs some editing. Suggestion: 'As predicted based on the known circuit (Figure 164 1C), this series of experiments indicates that inhibitory signals onto NS premotor neurons are transmitted to DE-3 motoneurons, thus limiting their firing frequency and counteracting their excitatory drive during crawling."

Changed as suggested.

Lines 86, 292 and 304 and Fig 4 legend: "Different from DE-3, In-Phase units showed a marked decrease in the maximum bFF along time." Suggestion: Replace the word "along" with 'across' time. Also replace those words in the Fig 4 legend and Line 80...."along" (replace with 'across') the different stages of crawling.

Changed as suggested.

Line 311: "bursts and a concurrent inhibitory input via NS (Figure 7). Coherent with this interpretation, the activity level of the Anti- Phase units was not influenced by these inhibitory signals". Suggestion: Replace the word "coherent" with 'consistent'.

Changed as suggested.

Line 332: "...offer the particular advantage of allowing electrical manipulation of individual neurons in wildtype adults," I am unsure what the authors are attempting to convey. Not sure what they mean by "wildtype" in this context and why that would matter.

“wildtype” was eliminated

We thank Reviewer #2 for the suggested edits to the text.